# Probability Paths
# and the Structure of Predictions over Time

**Zhiyuan Jerry Lin**[*]
Facebook
zylin@fb.com

**Hao Sheng**
Stanford University
haosheng@cs.stanford.edu

**Sharad Goel**
Harvard University
sgoel@hks.harvard.edu

## Abstract

In settings ranging from weather forecasts to political prognostications to financial projections, probability estimates of future binary outcomes often evolve over time. For example, the estimated likelihood of rain on a specific day changes by the hour as new information becomes available. Given a collection of such *probability paths*, we introduce a Bayesian framework—which we call the Gaussian latent information martingale, or GLIM—for modeling the structure of dynamic predictions over time. Suppose, for example, that the likelihood of rain in a week is 50%, and consider two hypothetical scenarios. In the first, one expects the forecast to be equally likely to become either 25% or 75% tomorrow; in the second, one expects the forecast to stay constant for the next several days. A time-sensitive decision-maker might select a course of action immediately in the latter scenario, but may postpone their decision in the former, knowing that new information is imminent. We model these trajectories by assuming predictions update according to a latent process of information flow, which is inferred from historical data. In contrast to general methods for time series analysis, this approach preserves important properties of probability paths such as the martingale structure and appropriate amount of volatility and better quantifies future uncertainties around probability paths. We show that GLIM outperforms three popular baseline methods, producing better estimated posterior probability path distributions measured by three different metrics. By elucidating the dynamic structure of predictions over time, we hope to help individuals make more informed choices.

## 1 Introduction

Probabilistic predictions of future binary outcomes are ubiquitous in real-world statistical and machine learning problems, ranging from election modeling [Erikson and Wlezien, 2012, Shirani-Mehr et al., 2018, Gelman et al., 2020] to weather forecasting [Esteves et al., 2019] to assessing mortality risk [Teres et al., 1987, Malmberg et al., 1997, Martinez-Alario et al., 1999, Yan et al., 2020] to risk assessment tools in criminal justice applications [Chouldechova, 2017, Lin et al., 2020]. Often such probabilistic predictions are not static, but rather evolve over time as more information becomes available. For instance, the estimated likelihood a candidate wins an election updates with every new poll that is conducted, and the estimated chance of rain on a given future day changes by the hour as new meteorological data become available.

In many of these domains, a large body of work aims to provide accurate, real-time forecasts that account for the very latest information. However, significantly less attention has been paid to understanding how the predictions themselves evolve over time. If there's a 35% chance of rain one week from now, what can we say about tomorrow's prediction of rain on that same day? In expectation, tomorrow's prediction must be the same as today's—if it were not, we would be better

---

[*]Research was conducted when the author was at Stanford University.

35th Conference on Neural Information Processing Systems (NeurIPS 2021).

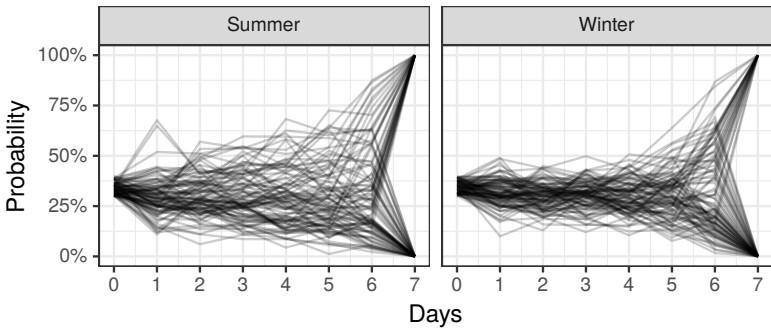

Figure 1: Dynamic forecasts of rainfall across several Australian cities during the summer and winter, starting one week before the target date. All forecasts initially indicate an approximately 35% chance of rain. But the forecast trajectories—which we call probability paths—have higher volatility in the summer than in the winter.

off updating today's prediction to match our expectation of tomorrow's forecast. In other words, such sequence of probabilistic predictions should be a martingale [Augenblick and Rabin, 2021, Foster and Stine, 2021, Gelman et al., 2020, Taleb et al., 2019, Taleb, 2018, Ely et al., 2015]. But aside from satisfying this martingale property, the forecast trajectories can vary widely from one setting to the next.

Consider, for example, the collection of dynamic weather forecasts presented in Figure 1. The plot shows the evolution of 7-day precipitation predictions for a set of Australian cities [Williams, 2011, Young and Young, 2018], where we have disaggregated the predictions by season (summer or winter) and restricted to those instances for which the 7-day forecast starts at approximately 35%. As is visually apparent, the mean prediction on any later day is still approximately 35%, as expected. But the summer forecasts are considerable more volatile than those in the winter. In the winter months (June–August), the prediction is unlikely to change substantially until immediately before the target date. In contrast, in the summer months (December–February), the prediction is likely to oscillate considerably—either up or down—with each passing day.

These patterns have immediate consequences for time-sensitive decision makers [Ferguson, 2004, Tsitsiklis and Van Roy, 1999, Chow and Robbins, 1963]. As a simple example, suppose one is planning a weekend picnic, with a preference both for sunshine and for sending out invitations as soon as possible. If one knows that a weather forecast is unlikely to change in the immediate future, one might opt to simply send out the invitations and hope for the best. Alternatively, knowing that more information is imminent—as reflected by forecasts that are likely to soon change—one might choose to wait another day before deciding to hold the party.

In this paper, we develop a Bayesian framework to investigate—and predict—the structure of dynamic probabilistic forecasts: the Gaussian latent information martingale, or GLIM. In contrast to existing techniques for time-series modeling, our approach is tailored to the specific properties of evolving probabilistic forecasts, which we call probability paths[2]. Most importantly, our approach satisfies the martingale property described above, with the expected value of future predictions along the path equal to the current prediction. Our model is trained directly on historical paths, and learns the heterogeneous, covariate-dependent structure of forecasts. As a result, it is rich enough to predict the type of structure illustrated in Figure 1. We show that this approach often outperforms existing state-of-the-art methods for quantifying uncertainty for evolving time series forecast, ostensibly because past general-purpose methods do not leverage the idiosyncratic properties of probability paths. As real-time forecasting becomes increasingly popular, we hope this work helps researchers and decision makers investigate and act on these dynamic predictions.

---

[2]Also known as stream of beliefs, streamflow forecast, or evolution of forecasts under different contexts.

## 2 Background

In this section, we formally define the problem we aim to address and introduce related literature pertaining to this topic.

### 2.1 Problem Setup

Suppose we are interested in a binary outcome $Y_T \in \{0, 1\}$ that will either occur or not at some fixed future time $T$ and the current time is $t = 0$. For example, we may be interested in whether it will rain $T = 7$ days from today. At time 0, we observe an initial probabilistic point estimate $Y_0 = y_0$ about the final outcome (e.g., from a static weather forecast model), as well as a collection of covariates $X$ (e.g., current and recent meteorological measurements). Our goal, then, is to accurately estimate how the predictions $\{Y\}_{t=1}^T$ of the event in question will evolve over time, described by a posterior distribution of $\{Y\}_{t=1}^T$ given all the information we have observed at $t = 0$.

While this problem might seem to be a straightforward time series modeling application at first sight, as shown by statisticians and economists [Augenblick and Rabin, 2021, Foster and Stine, 2021, Gelman et al., 2020, Taleb et al., 2019, Taleb, 2018, Ely et al., 2015], such evolving probabilistic predictions need to have an expectation equal to the initial prediction (i.e., being a martingale starting from $y_0$) and shall not display excessive or insufficient movement (i.e., appropriate amount of volatility as we will explain later in Section 4), putting additional constraints on the shape of predicted probability path distributions. Indeed, without any additional information, the only sensible posterior distributions of future predictions $\{Y\}_{t=1}^T$ are ones with the mean equal to $y_0$ [Gelman et al., 2020, Taleb, 2018], the current observed probabilistic point estimation.

It is worth noting that in this problem, unlike in many common time series modeling tasks, we are not interested in producing a sequence of point estimates for the future probability path $\{Y\}_{t=1}^T$, as a model that always outputs the current best probabilistic prediction $y_0$ for all future $\{Y\}_{t=1}^T$ is arguably the best point estimates we could possibly give without observing additional information. Instead, here we wish to provide an accurate estimate of the entire *posterior probability path distribution* conditioning on having observed all information available at $t = 0$. As we will explain in detail in Section 4, the quality of the estimated distribution can be assessed via examining some of the properties a probability path distribution should possess such as its posterior mean calibration, expected volatility, and credible interval coverage at different significance levels.

### 2.2 Related Work

In this subsection, we briefly review three classes of methods relevant to the problem of estimating the posterior distribution of probability paths $\{Y\}_{t=1}^T$.

**Martingale model of forecast evolution.** The model most relevant to our setting is probably the martingale model of forecast evolution, or MMFE, originally developed by Graves et al. [1986] and Heath and Jackson [1994] to estimate the evolution of production forecasts in supply chain optimization. In order to estimate how the forecasts of the production of a commodity at a future time $T$ evolve for $t = 0$ to $T$, learning from historical data, MMFE directly estimates the covariance matrix of increments between consecutive time points. With the learned covariance matrix, at $t = 0$ we are able to simulate arbitrary number of possible scenarios (hence the posterior distribution of forecast evolution) of how the production predictions for $t = T$ will be at time $t = 0, \ldots, T$. Although MMFE was originally invented a few decades ago, there are many recent successful applications and improvements of MMFE in a wide range of fields such as energy management [Xu et al., 2019], reservoir operation [Zhao et al., 2013, 2011], stock replacement [Boulaksil, 2016], and production planning [Albey et al., 2015]. Finally, recent work such as Augenblick and Rabin [2021] and Foster and Stine [2021] are studying martingale structures of probability paths, but they usually focus on developing diagnostic metrics for existing predictions instead of constructing predictive frameworks that capture probability path heterogeneity like GLIM.

**Probabilistic time series models.** When it comes to modeling time series data, probability path or not, arguably the most widely known class of models is probabilistic time series models. Famous examples include the autoregressive (AR), autoregressive moving average (ARMA), and autoregressive integrated moving average (ARIMA) models [Brockwell and Davis, 2016]. The autoregressive structure and the estimated Gaussian noise term allow us to simulate future time series evolution

through recursive sampling. If we additionally model the noise terms themselves as a function of noise terms from previous time steps, we have a *stochastic volatility model*. For example, when we assume the variance of the error terms—or innovation—follows a AR structure, we have a ARCH [Engle, 1982] model. Similarly, assuming a ARMA structure of error terms give the GARCH [Bollerslev, 1986] model. More recent development on stochastic volatility models include GARCHX [Sucarrat, 2020, Francq and Thieu, 2019, Engle and Patton, 2007], Augmented ARCH [Bera et al., 1992], and the Gaussian Copula Process Volatility (GCPV) model [Wilson and Ghahramani, 2010], which all allow for more complex structure of estimated variance over time and some of them grant the use of covariate information. Despite their vast success in numerous applications, most of these classical probabilistic time series models are primarily designed to model a single time series, rather than to model the multiple, independent time series in our motivating examples. Further, while some of these approaches may still be applied to model probability paths with some modifications, they are not designed to fully leverage the distinctive characteristics of probability paths, such as their martingale property, volatility level, or their finite time horizon (meaning the predictions must converge to 0 or 1 at a fixed time $T$).

**Bayesian deep sequence models.** Rapid development of deep learning models over the past decade has offered a new avenue of modeling time series data. Specifically, modern deep-learning based recurrent neural networks such as LSTM [Hochreiter and Schmidhuber, 1997, Goodfellow et al., 2016] and DeepAR [Salinas et al., 2019] can not only model and predict time series data, but also naturally incorporate the information of covariates into their predictions. There is a stream of recent work [Wen et al., 2017, Eisenach et al., 2020, Rangapuram et al., 2018] particularly focusing on providing better calibrated future time series predictions. Moreover, recent advancement in Bayesian deep learning has made uncertainty estimation possible in addition to point predictions through random regularization [Srivastava et al., 2014, Gal and Ghahramani, 2015, Kingma et al., 2015] or weights distribution [Blundell et al., 2015]. In spite of their significant model capacity, typical deep sequence models suffer from the same problem as classical probablistic time series models: they are not constrained by the martingale property in this bounded probability path setting. As we will show in Section 4, deep models tend to become overly conservative when it comes to uncertainty estimation and do not conform with the martingale requirement.

In summary, all of these techniques are powerful methods for modeling general time-series data, but none are tailored to the idiosyncratic properties of probability paths. In Section 4, we will evaluate GLIM against representative baseline models from each of the three classes of models and illustrate how GLIM can produce posterior future probability path samples with more accurate uncertainty estimation and better mean calibration.

## 3 Modeling Probability Paths

### 3.1 Gaussian latent information martingale

Now, we formally introduce the Gaussian latent information martingale, or GLIM, as a framework to model and predict probability paths. To model a probability path $\{Y\}_{t=0}^T$, we introduce a corresponding sequence of latent variables $\{Z\}_{t=1}^T$, where the scalar $Z_t$ intuitively represents the amount of information received between time $t-1$ and $t$. Receiving positive information (i.e., positive $Z_t$) makes the ultimate outcome $Y_T$ more likely, while receiving negative information makes the ultimate outcome less likely.

Now, we assume the latent variables $\{Z\}_{t=1}^T$ follow a multivariate Gaussian distribution with mean zero and $T \times T$ covariance matrix $\boldsymbol{\Sigma} = \boldsymbol{\Sigma}(X, \theta)$, where the covariance matrix can depend on both the observed covariates $X$ and a parameter $\theta$. Finally, we model $\{Y\}_{t=0}^T$ by assuming predictions along the probability path can be expressed in terms of the following conditional probabilities:

$$Y_t = \Pr\left(\gamma + \sum_{i=1}^T Z_i \geq 0 \mid Z_1, \ldots, Z_t\right), \tag{1}$$

where $\gamma$ is a fixed constant.

Eq. (1) is closely related to the latent variable formulation of logistic regression models, but we additionally condition on the latent variables. This expression has several attractive properties for modeling probability paths. First, $Y_t$ is naturally constrained to lie between 0 and 1. Second, $Y_T$ is

guaranteed to be either 0 or 1, since once we condition on $Z_1, \ldots, Z_T$, there is no randomness left in the expression. Finally, and most importantly, $Y_t$ satisfies the martingale property, as shown below.

**Proposition 1.** *For $\{Y\}_{t=0}^{T}$ satisfying Eq. (1), we have*

$$\mathbb{E}[Y_{t+1} \mid Y_0, \ldots, Y_t] = Y_t$$

*for $0 \leq t < T$.*

The proof is given by Appendix A.3 in the supplement material. Given a collection of probability paths, our goal is to infer the parameter $\theta$, as we describe next.

### 3.2 Model inference

We take a Bayesian approach to fit GLIM defined above. The main difficulty is deriving a tractable expression for the corresponding likelihood function. To do so, we first introduce some notation.

Given any $0 < t < T$, we divide the covariance matrix $\Sigma$ into four quadrants as follows:

$$\Sigma = \begin{bmatrix} \overbrace{\Sigma_{11}^t}^{t} & \overbrace{\Sigma_{12}^t}^{T-t} \\ \Sigma_{21}^t & \Sigma_{22}^t \end{bmatrix} \begin{matrix} \} & t \\ \} & T-t \end{matrix}$$

For $t = 0$, we further define $\Sigma_{22}^0 = \Sigma$. For an arbitrary matrix $\mathbf{M}$, we use the notation $\mathbf{M}_{(i,j)}$ to denote the $(i, j)$ entry in the matrix. Similarly, for an arbitrary vector $\mathbf{v}$, we use the notation $\mathbf{v}_{(i)}$ to denote the $i$-th entry in the vector.

Now, using this notation, Theorem 1 presents a recursive formula for computing the (log) probability density function for our Gaussian latent information martingale with parameter $\theta$. That expression can in turn be used to infer the parameters of the model from the observed data via maximum likelihood estimation or, alternatively, fully Bayesian inference.

**Theorem 1.** *Consider a probability path $y_0, \ldots, y_T$, with associated covariate vector $X$. For parameter $\theta$ and covariance matrix $\Sigma = \Sigma(X, \theta)$, suppose $Y_t$ is given by the GLIM model, defined in Eq. (1). Then there is a unique $\gamma$ such that $Y_0 = y_0$:*

$$\gamma = \Phi^{-1}(y_0)\sqrt{\sum_{i,j} \Sigma_{(i,j)}}, \tag{2}$$

*where $\Phi$ is the CDF of the standard normal distribution. Further, the log probability density function $f$ of the path $y_1, \ldots, y_T$ under the model is given by*

$$\log f(y_1, \ldots, y_T; \Sigma, \gamma) = -\sum_{t=1}^{T-1} \left( \log \tilde{\sigma}_t + \frac{(\Phi^{-1}(y_t) - \tilde{\mu}_t)^2}{2\tilde{\sigma}_t^2} - \frac{(\Phi^{-1}(y_t))^2}{2} \right)$$
$$+ y_T \cdot \log(y_{T-1}) + (1 - y_T) \cdot \log(1 - y_{T-1}),$$

*where $\tilde{\mu}_t$ and $\tilde{\sigma}_t$ are computed according to the following four-step procedure:*

1. *For $1 \leq t < T$, calculate $\Sigma^t$ and $\mathbf{a}^t$ according to the following expressions (and set $\Sigma^0 = \Sigma$):*

$$\Sigma^t = \Sigma_{22}^t - \Sigma_{21}^t (\Sigma_{11}^t)^{-1} \Sigma_{12}^t \qquad \mathbf{a}^t = \mathbf{1}^T \Sigma_{21}^t (\Sigma_{11}^t)^{-1}$$

2. *For $1 \leq t < T$, iteratively compute $z_t$:*

$$z_t = \frac{\sqrt{\sum_{i,j} \Sigma_{(i,j)}^t} \Phi^{-1}(y_t) - \sum_{i=1}^{t-1}(1 + \mathbf{a}_{(i)}^t)z_i - \gamma}{1 + \mathbf{a}_{(t)}^t}$$

3. *For $1 \leq t < T$, calculate $\mu^t$ (and set $\mu^0 = \mathbf{0}$):*

$$\mu^t = \Sigma_{21}^t (\Sigma_{11}^t)^{-1} [z_1, \cdots, z_t]^T$$

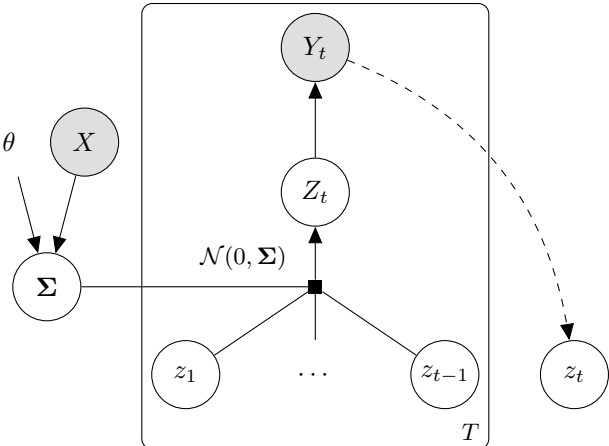

Figure 2: The directed graphical model under consideration of GLIM. Solid lines denote the generative model. Dashed lines denote the identification of $z_t$ from realized values of $Y_t$; this induction also requires knowledge of $\Sigma$ and $z_1, \cdots, z_{t-1}$, but those arrows are omitted in the diagram for simplicity. The parameters $\theta$ of the covariance matrix $\Sigma$ are learned from the data.

4. *Finally, for $1 \leq t < T$, calculate $\tilde{\mu}_t$ and $\tilde{\sigma}_t$*

$$\tilde{\mu}_t = \frac{\gamma + \sum_{i=1}^{t-1}(1 + \mathbf{a}_{(i)}^t)z_i + (1 + \mathbf{a}_{(t)}^t)\mu_{(1)}^{t-1}}{\sqrt{\sum_{i,j}\mathbf{\Sigma}_{(i,j)}^t}} \qquad \tilde{\sigma}_t = \frac{\sqrt{\mathbf{\Sigma}_{(1,1)}^{t-1}}(1 + \mathbf{a}_{(t)}^t)}{\sqrt{\sum_{i,j}\mathbf{\Sigma}_{(i,j)}^t}}.$$

Figure 2 shows a graphical representation of the model and inference process. A proof of Theorem 1 is given in Appendix A.1 of the supplementary material. The key observation is that given realized values of $Y_1, \ldots, Y_t$, together with the parameters of the underlying data-generating process, one can compute the implied values of the latent variables $Z_1, \ldots, Z_t$. Using change of variables, one can then compute the probability density function itself.

Much of the notational complexity in Theorem 1 stems from the fact that the latent variables can have a non-trivial correlation structure. This flexibility allows our model to better capture the patterns of real-world data. However, in some cases, it is sufficient to assume the latent variables are independent, which in turn considerably simplifies our expression of the density.

**Corollary 1.** *Suppose the latent variables $\{Z\}_{t=1}^T$ are independent, with a diagonal covariance matrix $\mathbf{\Sigma}$ having diagonal entries $\sigma_1^2, \sigma_2^2, \cdots, \sigma_T^2$. Then the expressions for $\tilde{\mu}_t$ and $\tilde{\sigma}_t$ in Theorem 1 have the following simplified form:*

$$\tilde{\mu}_t = \Phi^{-1}(y_{t-1})\sqrt{\frac{\sum_{i=t}^T \sigma_i^2}{\sum_{i=t+1}^T \sigma_i^2}}, \qquad \tilde{\sigma}_t = \frac{\sigma_t}{\sqrt{\sum_{i=t}^T \sigma_i^2}}.$$

A proof of Corollary 1 is given in Appendix A.2 of the supplementary material.

Given the expression for the likelihood derived in Theorem 1, there are multiple ways to carry out model inference. For example, one could sweep over the parameter space to maximize the likelihood of the observed probability paths under the model. Here we instead apply a Bayesian approach, putting a weakly informative prior on $\theta$, and then approximating its posterior distribution via Hamiltonian Monte Carlo (HMC), as implemented in `Stan` [Carpenter et al., 2017].

Without further constraints, $\theta$ is not fully identified by the data, since multiplying all of the latent variables in Eq. (1) by a positive constant does not affect the sign of the relevant expression. Thus, in our applications below, we constrain the scale of the latent variables by requiring $\text{Var}(Z_1) = \sigma_1^2 = 1$.

From a fitted GLIM model, it is straightforward to draw a probability path from the posterior distribution over paths. We do so in three steps. First, we draw a value of $\hat{\theta}$ from the inferred posterior.

Next, we draw the vector of latent variables $(z_1, \ldots, z_T)$ from the multivariate normal distribution $\mathcal{N}(\mathbf{0}, \mathbf{\Sigma}(X, \hat{\theta}))$. Finally, we compute values of $y_1, \ldots, y_T$ according to Eq. (1). As shown in the supplementary material, the value of Eq. (1) can be computed analytically, yielding:

$$y_t = \Phi \left( \frac{\gamma + \sum_{i=1}^{t-1} z_i + z_t + \sum_i \mu_{(i)}^t}{\sqrt{\sum_{i,\,j} \mathbf{\Sigma}_{(i,j)}^t}} \right) \tag{3}$$

for $1 \leq t < T$, where $\gamma$, $\mu^t$, and $\mathbf{\Sigma}^t$ are defined as in Theorem 1. For $t = T$, we have $Y_T = 1$ if and only if $\gamma + \sum_{i=1}^T z_i \geq 0$.

## 4 Experiments

We now explore the efficacy of GLIM through a series of experiments on two real-world datasets. We have additionally included a simulation study on a synthetic dataset in Appendix B to demonstrate GILM's empirical finite sample behavior. In all of our experiments, we use a covariance matrix $\mathbf{\Sigma}(X, \theta)$ with autoregressive structure and heteroskedastic variance. Specifically, we set

$$\mathbf{\Sigma}_{(i,j)} = \sigma_i \sigma_j \rho^{(n-|i-j|)}, \text{ where the variance at time } t \text{ is } \sigma_t^2 = \mathcal{G}_\beta(X, t).$$

In this setup, the covariance matrix $\mathbf{\Sigma}$ is parameterized by $\theta = (\rho, \beta)$. The first parameter, $\rho$, controls the correlation between latent variables, with $\rho = 0$ corresponding to independence. The second parameter, $\beta$, controls how the latent information evolves over time through the variance function $\mathcal{G}_\beta$. A simple example of the variance function is $\mathcal{G}_\beta(X, t) = \exp(\beta X(t - 1))$, where positive, zero, and negative values of $\beta^T X$ correspond to increasing, constant, and decreasing variance over time.

As we will demonstrate, this relatively simple structure using different variance functions often works well in practice. Figure 6 in Appendix B show that even with only two scalar parameters under this specification, GLIM is able to model a wide variety of probability path structures. However, other parameterization are also possible. For instance, deep kernel learning [Wilson et al., 2016] could be applied to facilitate more complex covariance matrix structure.

### 4.1 Experiment setup

We now turn to two more complex real-world prediction problems with evolving probability paths to assess the GLIM's performance: (1) the minute-by-minute win probability of professional basketball teams, updated as the game unfolds; and (2) forecasts of rainfall in Australian cities, updated daily as new information becomes available. In both prediction tasks, we observe the initial point prediction $y_0$ given by a pre-trained random forest predictor about the final outcome $Y_T$. The goal here is to accurately estimate the posterior probability path distribution, approximated by a sample of simulated posterior probability paths generated by the models to be evaluated. Figure 8a in Appendix D verifies that for both basketball and weather datasets, the probability paths themselves are generally well calibrated.

We compare GLIM against three representative baselines from each of the three aforementioned classes of models in Section 2: (1) MMFE: martingale method of forecast evolution [Heath and Jackson, 1994, Zhao et al., 2013]; (2) *LR*: a set of linear regression models $\{m_t\}$ that predict the future estimated probability at time $t$ [Brockwell and Davis, 2016]; and (3) *MQLSTM*: a Bayesian multi-horizon quantile LSTM model [Wen et al., 2017, Eisenach et al., 2020].

Assessing the quality of an estimated distribution is tricky: For each probability path in the dataset, we only observe one realized trajectory of the path, rendering metrics comparing two distributions (e.g., Kolmogorov–Smirnov statistic or cross-entropy) inappropriate.[3] Luckily, by taking advantage of the fact that we are modeling probability paths, we know there are a few key properties the true probability path distribution must satisfy and have accordingly devised three evaluation metrics to benchmark GLIM against the baseline models.

---

[3]Although it is possible to treat the observed path as a coarse approximation of ground truth distribution (i.e., one-point distribution), doing so will likely introduce an unacceptable level of noise in our evaluation.

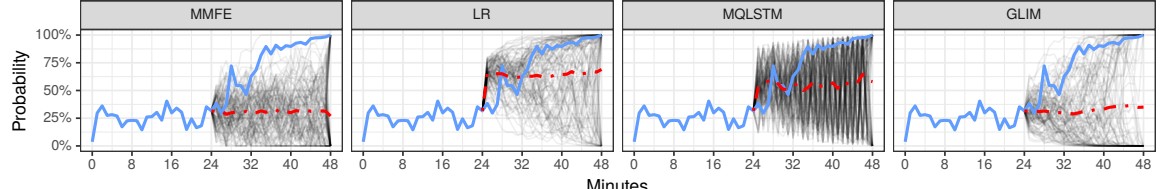

Figure 3: A comparison of real and predicted probability paths for a single basketball game, for GLIM and for our three baseline models: MMFE, LR, and MQLSTM. The blue lines are the observed ground-truth paths, the grey lines show 100 samples drawn from the distribution of paths predicted by each model, and the red lines show the mean predictions over time.

The first metric is the *posterior mean calibration mean squared error* (mean calibration MSE). Since the probability paths are expected to be martingales, posterior means of simulated paths should be around the initial prediction (i.e., $y_0$) for $t = 1, \ldots, T$ and we can empirically examine if the generated paths have a sample mean of $y_0$.

The second metric we check is the *expected volatility mean squared error* (volatility MSE). As proven by Augenblick and Rabin [2021] and Taleb [2018], for an evolving belief stream (i.e., a probability path), given an initial probabilistic prediction $y_0$, the accumulated squared increments, or volatility, defined as $Q_T = \sum_{i=1}^{T}(Y_i - Y_{i-1})^2$ should be equal to $y_0(1 - y_0)$ in expectation, i.e., $\mathbb{E}[Q_T] = y_0(1 - y_0)$. In practice, $\mathbb{E}[Q_T]$ can be estimated through averaging over the squared increment sum of a collection of sample paths.

Finally, if the realized paths are from the estimated probability path distribution, then for any $0 < t < T$, an empirical credible interval at significance level $\alpha$ is expected to cover exactly $\alpha$ proportion of observed paths. Specifically, we examine the *credible interval coverage error* (CI coverage error) at significance level $\alpha \in \{50\%, 80\%, 90\%, 95\%\}$ at the beginning, middle, and the end of probability paths for both the basketball and weather datasets.

In the experiments, for each model, all metrics are calculated using 100 simulated samples per probability path. All paths are simulated up until time $T - 1$, the the final end points are generated by drawing from Bernoulli($p = Y_{T-1}$) to ensure they converge to 1 or 0 at time $T$,

## 4.2 Basketball game outcome predictions

Suppose the second half of a basketball game is about to start. Based on all the available information, the probability that the home team wins is 60%. How is the game likely to evolve? Do we expect a game-deciding event to occur in the next few minutes? Can we safely run to the bathroom and not miss the action? More specifically, do we expect the 60% prediction to remain stable until the final minutes of the game, or, alternatively, do we expect it to quickly veer toward 0 or 100%?

Here we analyze regular-season NBA basketball games for the 2017–2018 and 2018–2019 seasons, training our model on the first season, and evaluating our predictions on the second[4]. To do so, we first created minute-by-minute probability paths for every 48-minute game. For every minute, we trained a separate random forest model to predict the final game outcome based on information available at that point, including: the current home-team and away-team score, the maximum score margin up until that point in the game, and the win rates for the home and away teams for the previous season. On the resulting probability paths, we fit our GLIM model—as well as our three baseline methods—to model predictions for the second half of the game[5]. All four models have access to the same information available at half-time, including the score, each team's performance in the previous season, and summary statistics for the probability path in the first half of the game (e.g., the minimum, maximum, average, and standard deviation of predictions). We use a regularized linear function $\mathcal{G}_\beta(X, t)$ for GLIM, which we describe in detail in Appendix C.1.

---

[4]Our basketball data were collected from `www.nba.com` with a third-party API client [Patel, 2018]. To avoid the problem of varying horizon lengths because of overtime periods, for any particular game, if the home team's score is greater or equal to the visiting team's, we mark the home team as the winner.

[5]For modeling purposes, we let $t = 0$ represent the 24th minute of the basketball game.

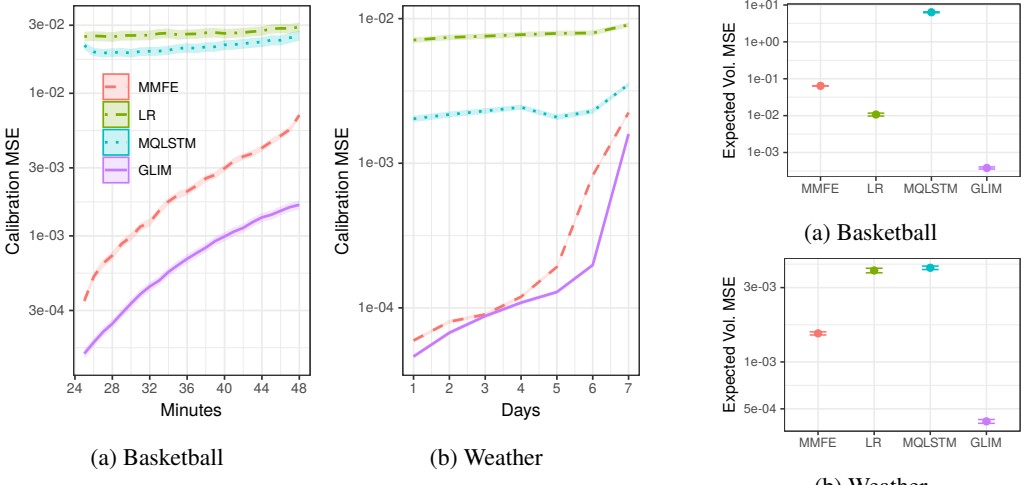

Figure 4: Mean calibration MSE, plotted on log scale. The estimated calibration MSE for GLIM is not identically zero because our estimate is based on a finite number of sample paths.

Figure 5: Volatility MSE, plotted on log scale.

In Figure 3, we show an illustrative example of a single game, modeled by all four methods. The identical blue curve in all four panels is the ground-truth probability path of the game. Starting at the 25th minute (i.e., at the start of the second half), each panel displays the distribution of probability paths predicted by each method, with the dashed red line indicating the mean prediction under each model over time. In this example, MQLSTM produces probability paths that have far too much movement showing significant oscillation between adjacent time steps. In contrast, LR produces paths that appear to appropriately fan out over time. Those paths, however, skew towards the positive outcome just like MQLSTM paths, failing to satisfy the martingale property, as indicated by the red line. MMFE is the only baseline model that is able to maintain the posterior mean being centered around $y_0$. Unfortunately, all its paths are generally concentrated around $y_0$ until the end and have assigned pretty small density around the ground-truth blue path. Finally, GLIM generates predicted paths that are both calibrated (as is theoretically guaranteed) and that appear to appropriately capture the path volatility. One interpretation for GLIM's simulated paths is that given all information we have seen at the 24th minute and the point prediction $y_0$ being around 30%, if the home team is going to lose the game, we can expect to have a confident prediction at around the 36th minute; but if the home team is going to win, we might not be so sure about it until the very end of the game.

| Min. | $\alpha$ | MMFE | LR | MQLSTM | GLIM |
|------|------|------|------|--------|------|
| 25 | 50% | 0.13 | -0.32 | **-0.08** | **0.07** |
|    | 80% | **0.04** | -0.46 | -0.10 | **0.03** |
|    | 90% | **0.00** | -0.46 | -0.10 | **0.01** |
|    | 95% | **-0.02** | -0.45 | -0.10 | **-0.01** |
| 36 | 50% | **0.03** | -0.06 | -0.09 | 0.07 |
|    | 80% | **-0.03** | -0.08 | -0.12 | 0.05 |
|    | 90% | -0.05 | -0.08 | -0.12 | **0.02** |
|    | 95% | -0.05 | -0.07 | -0.12 | **0.01** |
| 47 | 50% | **-0.01** | **0.02** | -0.14 | **0.01** |
|    | 80% | -0.06 | **0.01** | -0.22 | **0.02** |
|    | 90% | -0.06 | **-0.01** | -0.24 | **0.03** |
|    | 95% | -0.05 | **-0.02** | -0.24 | **0.02** |

(a) Basketball $\alpha$-level CI coverage errors

| Day | $\alpha$ | MMFE | LR | MQLSTM | GLIM |
|------|------|------|------|--------|------|
| 1 | 50% | 0.06 | -0.14 | **-0.01** | **0.01** |
|   | 80% | **0.03** | -0.19 | **-0.02** | **-0.01** |
|   | 90% | **0.00** | -0.19 | -0.03 | -0.03 |
|   | 95% | **0.00** | -0.17 | -0.03 | -0.03 |
| 3 | 50% | **0.03** | -0.09 | **-0.02** | **0.03** |
|   | 80% | **0.00** | -0.13 | -0.03 | **0.01** |
|   | 90% | **-0.01** | -0.12 | -0.04 | **0.00** |
|   | 95% | **-0.02** | -0.11 | -0.05 | **-0.02** |
| 6 | 50% | 0.15 | 0.21 | 0.15 | **0.01** |
|   | 80% | 0.10 | 0.13 | 0.11 | **-0.01** |
|   | 90% | 0.05 | 0.07 | 0.06 | **-0.01** |
|   | 95% | **0.02** | **0.03** | **0.03** | **-0.02** |

(b) Weather $\alpha$-level CI coverage errors

Table 1: Credible interval coverage at significance level $\alpha$ at different time points. Positive values indicate overly wide credible intervals, and negative values suggest overly narrow credible intervals. Standard errors are generally at or below 0.01 level and are omitted from this table. The lowest coverage errors (or within $\pm 0.02$ from the lowest absolute error) are in bold. GLIM's performance is consistently the best or indistinguishable from the best in nearly all (time, $\alpha$) combinations.

Moving beyond this single example, Figure 4a, Figure 5a, and Table 1a show the mean calibration MSE, expected volatility MSE, and credible interval coverage respectively. As displayed in the plot, GLIM outperforms all three baselines across the board. Particularly, Figure 4a are Figure 5a are plotted in log scale, suggesting GLIM is outperforming baselines by a few orders of magnitudes on those metrics.

### 4.3 Weather predictions

Finally, we consider the problem of modeling dynamic forecasts of precipitation. Whereas basketball games may ostensibly be well-modeled as a biased random walk, weather outcomes arguably have much more complicated temporal dynamics [Palmer, 1991]. Specifically, we use a dataset of Australian rainfall observations [Williams, 2011, Young and Young, 2018], and construct daily predictions starting seven days in advance of the target date. Here, starting at day zero, we aim to model the evolution of our predictions on whether it will rain on the target date. In this case, our predictions were based on a variety of meteorological features, including air pressure, wind speed, location, and month of year. We randomly sampled 10,000 target dates in the dataset prior to 2014 for training our models, and randomly sampled 10,000 target dates in or after 2014 for testing. GLIM and our baseline models were then fit to the training paths, using the same features as above. We set $\rho = 0$ in this case and use a regularized quadratic function $\mathcal{G}_\beta(X, t)$ for GLIM, which we describe in detail in Appendix C.2.

GLIM and three baseline models' mean calibration MSE, expected volatility MSE, and credible interval coverage for the weather dataset are displayed in Figure 4b, Figure 5b, and Table 1b respectively. Similar to what we observe in the basketball dataset, GLIM performs better than all three baseline models. In summary, the results from both the basketball game outcome predictions and weather predictions datasets suggest that GLIM is a desirable framework for modeling probability paths both in theory and in practice.

## 5 Conclusion

In this paper, we formally investigated the structure of probabilistic predictions that evolve over time. In doing so, we introduced the Gaussian latent information martingale (GLIM), a Bayesian framework to model these probability paths. In contrast to previous work in this area, GLIM can naturally incorporate covariates, is able to produce multi-step predictions without explicitly modeling changes in covariates over time, and, perhaps most importantly, preserves important properties of probability paths such as the martingale structure and amount of movements. We investigated GLIM's performance on two real-world datasets in addition to a synthetic dataset (Appendix B), helping us understand the dynamic structure of predictions over time.

Aside from being an intriguing problem in and of itself, understanding the structure of evolving probability paths can aid time-sensitive decision makers choose an appropriate action [Ferguson, 2004, Tsitsiklis and Van Roy, 1999, Chow and Robbins, 1963]. Examples range from helping ordinary individuals plan weather-contingent events, to physicians treating patients with rapidly changing symptoms [Wassenaar et al., 2015], to prosecutors making judgements based on updating information [Lin et al., 2019].

With the increasing availability of real-time data across domains, rapidly evolving forecasts are also likely to become more common. GLIM offers one promising route for explicitly modeling—and, in turn, predicting—the trajectory of forecasts. Looking forward, we hope our work sparks further interest in uncovering and leveraging the subtle structure of such dynamic predictions.

## Acknowledgements

We thank Johann Gaebler, Daniel Jiang, Zhen Liu, Sabina Tomkins, and Yuchen Xie for helpful conversations throughout the project, and Nicolas Lambert and Jessica Su for in-depth discussions on a preliminary version of this work. Code to replicate our experiments is available online at: `https://github.com/ItsMrLin/probability-paths`.

**Funding support:** The authors acknowledge that they received no funding in support of this research.
**Competing interests:** The authors declare that they have no competing interests.

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
