# A  Proofs

## A.1  Proof of Theorem 1

*Proof.* Initially, by definition of $Y_0$, the observed $y_0$ can be expressed as

$$y_0 = \Phi\left(\frac{\gamma + \mathbb{E}[\sum_{i=1}^{T} Z_i]}{\sqrt{\text{Var}(\sum_{i=1}^{T} Z_i)}}\right) = \Phi\left(\frac{\gamma}{\sqrt{\sum_{i,j} \boldsymbol{\Sigma}_{(i,j)}}}\right)$$

and $\gamma$ can be uniquely identified as

$$\gamma = \Phi^{-1}(y_0)\sqrt{\sum_{i,j} \boldsymbol{\Sigma}_{(i,j)}}.$$

Similarly, definition of $Y_t$ gives

$$Y_t = 1 - \Phi\left(\frac{-\gamma - \sum_{i=1}^{t-1} Z_i - Z_t - \bar{\mu}_t}{\bar{\sigma}_t}\right).$$

By symmetry, we have

$$Y_t = \Phi\left(\frac{\gamma + \sum_{i=1}^{t-1} Z_i + Z_t + \bar{\mu}_t}{\bar{\sigma}_t}\right)$$

$$\Phi^{-1}(Y_t) = \frac{\gamma + \sum_{i=1}^{t-1} Z_i + Z_t + \bar{\mu}_t}{\bar{\sigma}_t}, \tag{4}$$

where $\Phi(\cdot)$ is the standard normal cumulative density function (CDF); $\bar{\mu}_t$ and $\bar{\sigma}_t$ are respectively the mean and standard deviation of the Gaussian variable $\sum_{i=t+1}^{T} Z_i$ conditioned on previous latent information variable values.

Now we show how to obtain the defining parameters $\bar{\mu}_t$ and $\bar{\sigma}_t$ for the conditional distribution of $(\sum_{i=t+1}^{T} Z_i \mid Z_1 = z_1, \dots, Z_t = z_t)$.

At time $t$, given the realization of the latent information variables $z_1, \dots, z_t$, the remaining ones will follow a multivariate Gaussian distribution:

$$(Z_{t+1}, \dots, Z_T \mid Z_1 = z_1, \dots, Z_t = z_t) \sim \mathcal{N}(\mu^t, \boldsymbol{\Sigma}^t) \tag{5}$$

with

$$\mu^t = \boldsymbol{\Sigma}_{21}^t \left(\boldsymbol{\Sigma}_{11}^t\right)^{-1} [z_1, \dots, z_t]^T$$

$$\boldsymbol{\Sigma}^t = \boldsymbol{\Sigma}_{22}^t - \boldsymbol{\Sigma}_{21}^t \left(\boldsymbol{\Sigma}_{11}^t\right)^{-1} \boldsymbol{\Sigma}_{12}^t.$$

The terms $\mu^t$ and $\boldsymbol{\Sigma}^t$ are simply the conditional mean and variance of the multivariate Gaussian distribution $\mathcal{N}(\mathbf{0}, \boldsymbol{\Sigma})$ when conditioned on the first $t$ latent information variables.

Then the conditional sum $(\sum_{i=t+1}^{T} Z_i \mid Z_1 = z_1, \dots, Z_t = z_t)$ will follow a Gaussian distribution $\mathcal{N}(\bar{\mu}_t, \bar{\sigma}_t^2)$. Let $\mathbf{a}^t$ be $\mathbf{1}^T \boldsymbol{\Sigma}_{21}^t \left(\boldsymbol{\Sigma}_{11}^t\right)^{-1}$, we have mean $\bar{\mu}_t$ and variance $\bar{\sigma}_t^2$ being

$$\bar{\mu}_t = \mathbf{1}^T \mu^t$$

$$= \mathbf{1}^T \boldsymbol{\Sigma}_{21}^t \left(\boldsymbol{\Sigma}_{11}^t\right)^{-1} [z_1, \dots, z_t]^T$$

$$= \mathbf{a}^t [z_1, \dots, z_t]^T = \sum_{i=1}^{t} \mathbf{a}_{(i)}^t z_i$$

$$\bar{\sigma}_t^2 = \sum_{i,\,j} \boldsymbol{\Sigma}_{(i,j)}^t,$$

where $\boldsymbol{\Sigma}_{(i,j)}^t$ is the element at the $i$-th row, $j$-th column in $\boldsymbol{\Sigma}^t$.

Once we have observed $y_t$ and identified $z_1, \ldots, z_{t-1}$, by substituting and rearranging Eq. (4), we can uniquely identify $Z_t$:

$$Z_t = z_t = \frac{\bar{\sigma}_t \Phi^{-1}(y_t) - \sum_{i=1}^{t-1}(1 + \mathbf{a}_{(i)}^t)z_i - \gamma}{1 + \mathbf{a}_{(t)}^t}.$$

Therefore, given $\mathbf{\Sigma}$ and $\gamma$, once we have observed $y_1, \ldots, y_t$, we can uniquely identify $z_1, \ldots, z_t$. As a result, $(\Phi^{-1}(Y_t) \mid Y_{t-1} = y_{t-1}, \ldots, Y_1 = y_1)$ is equivalent of $(\Phi^{-1}(Y_t) \mid Z_{t-1} = z_{t-1}, \ldots, Z_1 = z_1)$.

Now we are going to find the conditional distribution for $(\Phi^{-1}(Y_t) \mid Z_{t-1} = z_{t-1}, \ldots, Z_1 = z_1)$.

As shown in Eq. (5), when conditioning on the first $t$ latent information variables $Z$s, the remaining ones follow distribution $\mathcal{N}(\mu^t, \mathbf{\Sigma}^t)$. Then when conditioned on $(Z_{t-1} = z_{t-1}, \ldots, Z_1 = z_1)$, the mean and variance of the conditional marginal distribution of $Z_t$ are simply the first elements in $\mu^{t-1}$ and $\mathbf{\Sigma}^{t-1}$, namely:

$$(Z_t \mid Z_{t-1} = z_{t-1}, \ldots, Z_1 = z_1) \sim \mathcal{N}(\mu_{(1)}^{t-1}, \mathbf{\Sigma}_{(1,1)}^{t-1}).$$

Substituting $\bar{\mu}_t$, $\bar{\sigma}_t^2$, and replacing the conditioned $Z_{t-1}, \ldots, Z_1$ with $z_{t-1}, \ldots, z_1$ in Eq. (4), we can obtain the conditional distribution of $\Phi^{-1}(Y_t)$, which is a linear transformation of $(Z_t \mid Z_{t-1} = z_{t-1}, \ldots, Z_1 = z_1)$:

$$(\Phi^{-1}(Y_t) \mid Z_{t-1} = z_{t-1}, \ldots, Z_1 = z_1) \sim \mathcal{N}(\tilde{\mu}_t, \tilde{\sigma}_t^2),$$

with

$$\tilde{\mu}_t = \frac{\gamma + \sum_{i=1}^{t-1}(1 + \mathbf{a}_{(i)}^t)z_i + (1 + \mathbf{a}_{(t)}^t)\mu_{(1)}^{t-1}}{\bar{\sigma}_t}$$

$$\tilde{\sigma}_t = \frac{\sqrt{\mathbf{\Sigma}_{(1,1)}^{t-1}}(1 + \mathbf{a}_{(t)}^t)}{\bar{\sigma}_t}.$$

Finally, by applying change-of-variable trick, we are able to write out the conditional likelihood $P(Y_t = y_t \mid y_{t-1}, \ldots y_1; \mathbf{\Sigma}, \gamma)$ in terms of $\Phi^{-1}(y_t)$ for $0 < t < T$ as

$$P(Y_t = y_t \mid y_{t-1}, \ldots y_1; \mathbf{\Sigma}, \gamma)$$
$$= P(Y_t = y_t \mid z_{t-1}, \ldots z_1; \mathbf{\Sigma}, \gamma)$$
$$= P(\Phi^{-1}(Y_t) = \Phi^{-1}(y_t) \mid z_{t-1}, \ldots z_1; \mathbf{\Sigma}, \gamma) \times \left| \frac{\partial \Phi^{-1}(y)}{\partial y} \right|_{y=y_t} \right|$$
$$= \frac{\varphi(\Phi^{-1}(y_t); \tilde{\mu}_t, \tilde{\sigma}_t^2)}{\varphi(\Phi^{-1}(y_t))}, \tag{6}$$

where $\varphi(\cdot; \tilde{\mu}_t, \tilde{\sigma}_t^2)$ is the PDF of a normal distribution with mean $\tilde{\mu}_t$ and variance $\tilde{\sigma}_t^2$ and $\varphi(\cdot)$ is the standard normal PDF. When $t = T$, we have $P(Y_T = y_T \mid y_{T-1}, \ldots y_1; \mathbf{\Sigma}, \gamma) = P(Y_T = y_T \mid y_{T-1}) = y_{T-1}^{y_T}(1 - y_{T-1})^{(1-y_T)}$ as it follows a Bernoulli distribution with $p = y_{T-1}$ by definition. Multiplying $P(Y_t = y_t \mid y_{t-1}, \ldots y_1; \mathbf{\Sigma}, \gamma)$ for all $0 < t \le T$ gives the likelihood of the path $y_1, \ldots, y_T$. With expansion of normal PDFs, log transformation, and some rearrangement of constants, we have the expression for log PDF of $y_1, \ldots, y_T$ as presented in Theorem 1. $\square$

## A.2 Proof of Corollary 1

*Proof.* If $\mathbf{\Sigma}$ is diagonal, we have $\mathbf{\Sigma}_{21}^t = \mathbf{\Sigma}_{12}^t = \mathbf{0}$ for $t = 1, \ldots, T$, indicating $\mathbf{\Sigma}^t = \mathbf{\Sigma}_{22}^t$, $\mu_t = \mathbf{0}$, and $\mathbf{a}^t = \mathbf{0}$.

We have,

$$z_t = \frac{\bar{\sigma}_t \Phi^{-1}(y_t) - \sum_{i=1}^{t-1}(1 + \mathbf{a}_{(i)}^t)z_i - \gamma}{1 + \mathbf{a}_{(t)}^t}$$

$$= \sqrt{\sum_{i=t+1}^{T} \sigma_i^2} \Phi^{-1}(y_t) - \sum_{i=1}^{t-1} z_i - \gamma.$$

Hence,

$$\gamma + \sum_{i=1}^{t} z_i = \sqrt{\sum_{i=t+1}^{T} \sigma_i^2} \, \Phi^{-1}(y_t).$$

Then

$$\tilde{\mu}_t = \frac{\gamma + \sum_{i=1}^{t-1} z_i}{\sqrt{\sum_{i=t+1}^{T} \sigma_i^2}} = \Phi^{-1}(y_{t-1}) \sqrt{\frac{\sum_{i=t}^{T} \sigma_i^2}{\sum_{i=t+1}^{T} \sigma_i^2}}$$

$$\tilde{\sigma}_t = \frac{\sqrt{\boldsymbol{\Sigma}_{(1,1)}^{t-1}}}{\sqrt{\sum_{i,j} \boldsymbol{\Sigma}_{(i,j)}^{t}}} = \frac{\sigma_t}{\sqrt{\sum_{i=t}^{T} \sigma_i^2}}.$$

$\square$

## A.3 Proof of Proposition 1

*Proof.* The proof follows by repeatedly applying the law of iterated expectations. In particular,

$$
\begin{aligned}
& \mathbb{E}[Y_{t+1} \mid Y_0, \ldots, Y_t] \\
&= \mathbb{E}[\mathbb{E}[Y_{t+1} \mid Z_1, \ldots, Z_t] \mid Y_0, \ldots, Y_t] \\
&= \mathbb{E}[\mathbb{E}[\mathbb{E}[Y_T \mid Z_1, \ldots, Z_{t+1}] \mid Z_1, \ldots, Z_t] \mid Y_0, \ldots, Y_t] \\
&= \mathbb{E}[\mathbb{E}[Y_T \mid Z_1, \ldots, Z_t] \mid Y_0, \ldots, Y_t] \\
&= \mathbb{E}[Y_t \mid Y_0, \ldots, Y_t] \\
&= Y_t.
\end{aligned}
$$

$\square$

# B Synthetic Data Simulation Study

In addition to the real-world datasets we have studied, we also examine the expressiveness of GLIM by generating probability paths under the model with different parameter settings. In particular, we consider paths of length $T = 10$, with a constant covariate $X = 1$, initial value $y_0 = 0.75$, and $\mathcal{G}_\beta(X, t) = \exp(\beta X(t - 1))$. Figure 6 shows the path distribution for several different combinations of $\rho$ and $\beta$ values. As is visually apparent from the figure, even with such simple parameterization of the covariance matrix, the model can produce paths exhibiting a wide range of structures while maintaining the martingale property. For example, one can generate paths exhibiting variance near the end of the time period ($\beta = 1$), or near the beginning ($\beta = -1$).

Under quite general regularity conditions, Bayesian posterior means yield consistent parameter estimates [Miller, 2018], yet their finite-sample properties are not always as nice. Here, we explore the efficacy of GLIM to recover estimates in a limited data setting.

In our simulated setting, we consider a time horizon of $T = 5$ steps, with probability paths associated with a single binary covariate $X$. We tested $5 \times 5 = 25$ different pairs of $\beta$ and $\rho$ values, ranging from -0.4 to 0.4. For each $(\beta, \rho)$ pair, we generated 50 synthetic datasets, with each dataset comprised of 500 probability paths, and half of the paths having $X = 0$ and the other having $X = 1$. On each synthetic dataset, we fit a GLIM model with MCMC to compute the posterior means $\hat{\beta}$ and $\hat{\rho}$ of the model parameters.

We plot the results of this exercise in Figure 7. For each parameter choice, we plot the mean and standard deviation of $\hat{\beta}$ and $\hat{\rho}$ across the 50 synthetic datasets. For all choices of $\beta$ and $\rho$, the posterior means are tightly clustered around the true values, indicating our inference is generally working well, even with short paths and relatively small datasets. Estimates are somewhat more dispersed when the correlation across latent variables is higher (e.g., when $\rho = 0.4$), ostensibly because we shrink the effective number of sample points in this case, creating a more challenging inference problem. Nevertheless, these results suggest GLIM is often able to effectively recover model parameters.

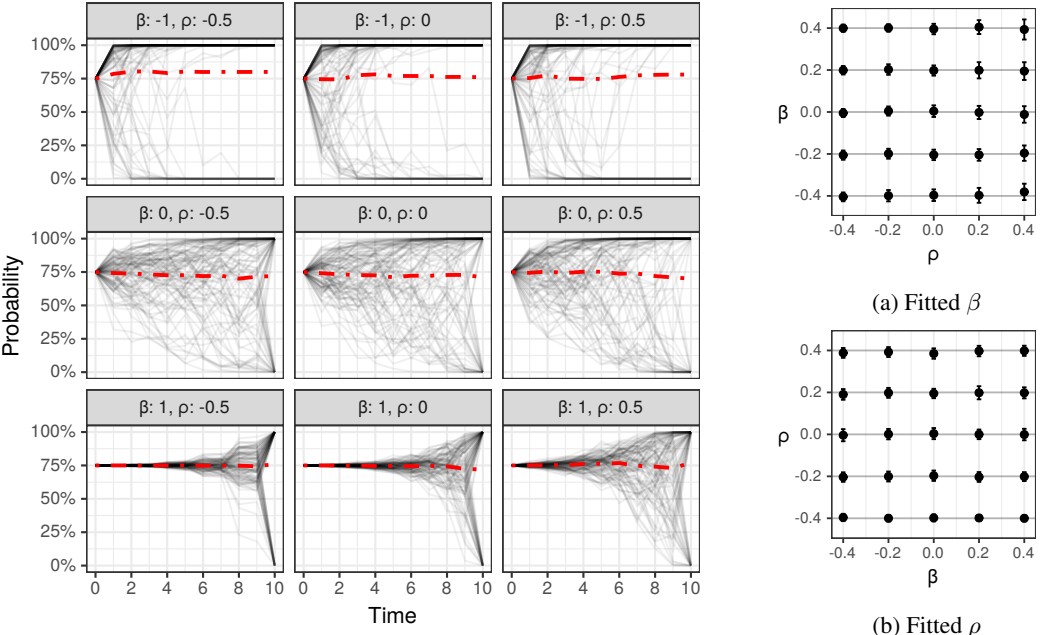

Figure 6: The distribution of probability paths for different co-variance structures of the latent variables, parameterized by $\beta$ and $\rho$. The grey lines show 100 sample probability paths, and the red lines show the path sample means over time.

(a) Fitted $\beta$

(b) Fitted $\rho$

Figure 7: The distribution of inferred parameters across synthetic datasets, for different choices of $\rho$ and $\beta$.

## C   Variance Function Specifications

### C.1   Basketball variance function

For the basketball dataset, we set $\mathcal{G}_\beta(X, t) = \exp(c \cdot \text{sigmoid}(\beta X)(t - 1))$ where $c$ is a fixed scaling constant to limit the maximum possible value of variance. We set $c$ to be $\frac{5}{T-1} = \frac{5}{23}$ for the basketball dataset. This structure implies the minimum diagonal variance should be have an increasing structure overtime as suggested by analysis in Foster and Stine [2021] and should be between 1 and $exp(5) \approx 148$, preventing extreme values that might be introduced by the exponential function.

### C.2   Weather variance function

For the weather dataset, we set $\mathcal{G}_{\beta,p}(X, t) = \exp(a(t - 1)^2 + b(t - 1) + c_t)$, where $a = \text{softplus}(\beta X) = \log(1 + \exp(\beta X))$, $b = -pa$. $p$ is constrained to be between 4 and 5, and $\mathbf{c} = [c_1, \ldots, c_T] = [-0.7, -1.5, -1.5, -1.5, -1.5, -1.5, -0.3, 2]$ is a fixed constant intercept vector determined empirically. This formulation is used because we notice a significant variance of those path are concentrating around the last step during the analysis of the weather probability paths. As a result, we use $\mathbf{c}$ to regularize the overall shape of the variance function and let the learned $\beta$ and $p$ parameter to account for idiosyncratic structure for each individual path.

## D   Outcome Prediction Model Calibration

Figure 8 shows the calibration plots for the predictions we observe in the basketball and weather prediction datasets. In both datasets, observed predictions are generally well-calibrated.

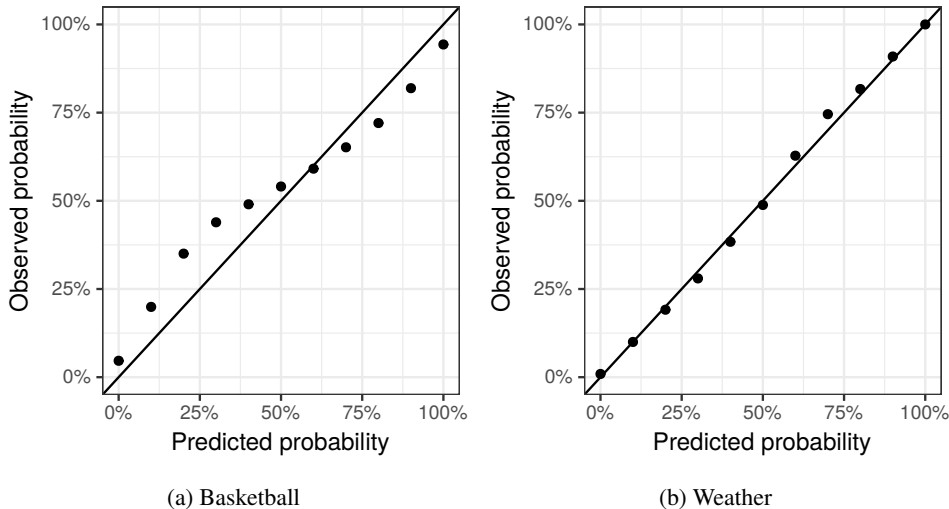

(a) Basketball

(b) Weather

Figure 8: Outcome model calibration plots. Ground truth probability paths are generally well calibrated for both basketball and weather datasets.