# OpenReview forum: "Probability Paths and the Structure of Predictions over Time"
_NeurIPS.cc/2021/Conference — NeurIPS 2021 Poster_

### Official Review · Reviewer_oVRK · 2021-07-03

**Rating:** 5
**Confidence:** 4

**Summary:**

The paper presents a latent Gaussian model for the progression of probability forecasts. Under this model, forecasts are a martingale and the paper describes a sampling strategy to predict the trajectory of the remaining (T - t) forecasts based on the first t observed forecasts. The paper applies the method on real world data and the results are promising.

-----
Update after author feedback period:
I appreciate the authors' responses to other referee's comments. After going through the responses and subsequent discussion, I unfortunately must lower my score. While the model is interesting, I do not believe the relevant empirical comparisons have been done. Namely, the methods of Foster & Stine can serve as a way to "post-process" a probability path to improve the forecast accuracy (measured, e.g., by MSE). I think a comparison to post-processed procedures is vitally necessary here and, in particular, using MSE. At present it is not clear to me that the proposed model will outperform these straw-men.

**Limitations And Societal Impact:**

Not addressed

**Main Review:**

Originality: The latent Gaussian model seems clever (though I have some clarifying questions, see below) and the idea of modeling trajectories of forecasts is a nice one.

Quality: The model and inferential approach seems sound. However, the specific form of the model seems a bit ad hoc. Can you provide more motivation / intuition for the additive term $\sum_{t = 1}^{T}{Z_{t}}$ in Equation (1)? One might reasonably consider an alternative model of the form $Y_{t} = P(\gamma + Z_{t})$ where the $Z_{t}$'s are dependent (e.g. the covariance matrix of $Z$ has a first-order autoregressive structure). As presented the model seems a bit ad hoc / under-motivated and I think the paper would be considerably strengthened by arguing why some "straw man" alternatives are inadequate (e.g. do not satisfy the martingale property or have some other disqualifying characteristic).

Semi-relatedly, how sensitive are your results to the choice of variance function? The elicitation in Appendix D for the basketball dataset is a good start but it would be useful to, at a minimum, comment in the main text on the sensitivity to the choice of variance function and give interested practitioners some guidance for choosing this function.

Clarity: On the whole, the paper is very well-written and easy to follow.

Significance: I think the basic model presented here is quite interested and will stimulate new research interests.

**Time Spent Reviewing:**

3

---

### Official Review · Reviewer_tNwH · 2021-07-15

**Rating:** 5
**Confidence:** 4

**Summary:**

The paper titled "Probability Paths and the Structure of Predictions over Time" proposes and analyses a probabilistic model called Gaussian Latent Information Martingale (GLIM) to model probability paths, a finite sequence $\\{Y_t\\}_{t \in \\{0,1,\ldots,T\\}}$ of probabilities for an event happening at time $T$ with $Y_t$ being the best guess of the probability of the event at time $t$ conditional on information available till that time. After proposing the generative model which satisfies certain desirable properties like martingale property and $Y_T$ being either $0$ or $1$, the authors discuss an algorithm to compute the likelihood and thereby allowing inference. This is followed by experiments which range from doing inference on a synthetic data set for sanity check to comparing GLIM against three other models on basketball game outcome and rain prediction data.

**Limitations And Societal Impact:**

Not relevant.

**Main Review:**

This is a well-written paper discussing a very interesting problem complete with motivation, technical analysis and experiments. The problem of modeling probability paths seems so obviously important in retrospect, and yet, I am embarrassed to admit, I had never thought about it or seen it discussed before. The latent variable model, GLIM, is a very simple and elegant yet powerful model that has been proposed by the authors to solve this very problem. This model can be easily extended to a more complicated one by adding various bells and whistles, as also acknowledged by the authors.

One shortcoming is that the paper fails to discuss why those specific $\mathcal{G}_\beta(X,t)$ functions were used for the experiments. They seem very ad-hoc and could potentially decide whether the model works well or not.

EDIT: After extensive reviewer discussions I am decreasing my rating of the paper from a clear accept to a weak reject. My earlier positive review was based on incomplete understanding of the subject of forecasting. The main point because of which I cannot accept this paper is that martingality in and of itself is not the goal of a forecasting model. The three metrics used in the paper fail to be relevant for forecast accuracy, something like Brier score would have been better. The model proposed, though elegant, may be sacrificing accuracy and without proper metrics we wouldn't know. In light of this it isn't clear what the goal of the paper is anymore. This fatal flaw along with the fact that the authors failed to engage in discussions expect with one reviewer (j4Fr) brings the paper into the territory of reject.

**Time Spent Reviewing:**

6 hours

---

### Official Review · Reviewer_j4Fr · 2021-07-16

**Rating:** 5
**Confidence:** 4

**Summary:**

The main contributions of this paper are proposing a Bayesian model that satisfies the martingale property point forecasts should have:

- This paper provides a nice cool idea to make Bayesian models satisfy the martingale property. Eqn (1) is a cool idea -
- Its a pretty nice comparison across datasets - its nice to actually track this
- The metrics actually track something useful (like Talebs/AugRab's martingale trade). This converges to the volatility of a diffusion (as shown by Taleb later - https://d1wqtxts1xzle7.cloudfront.net/60195149/clayton3-with-cover-page-v2.pdf?Expires=1626415145&Signature=d~fkCG~WwZ14pZopiG5MwXC~1Dbt4y9C2w-X1GQeKvFJhwpPoiKg0wV9DIN5XrdWEVbY8QGTLTzxcFRVJ3URwCSFeMQjCrapJ3BKf5m7fx7g0Q4X2FpLhfFfqTA6rse3oW8AOADxHDm7yK0-9bE6l3cyDZ2xjI43FMEcoZzWqXoBGN6RS1KnqmbVwXf4SiT6~zItizMdMGo0wYOUIKtei6-zSZ8cM8BP0eOtTAaqy~pnRo11yqXR0vjQWN0uZERfyFmHnBGIXo4vGnihTPbOs5SOnyZabMlbiqcTT8rc7sBV4N3GnwYYF5zeX4w8KW9FELrcPTvqFKVzOm5axR2aTA__&Key-Pair-Id=APKAJLOHF5GGSLRBV4ZA)



**Limitations And Societal Impact:**

1) I think overall the model fails to address the primary ideas of the martingale property elucidated by Foster and Stine and Taleb et al. The martingale property is fundamentally about a more accurate forecast
2) To that end comparing against fair baselines (MQTransformer) and other models to which the Foster and Stine Martingale Filter have been applied are more relevant
3) This claim about Foster and Stine ("but they usually focus on developing diagnostic metrics instead of building predictive frameworks that capture probability path heterogeneity like GLIM") I think is wrong. Foster and Stine seem to formalize the connection to why we care about the martingale property at all (it like calibration guarantees a >= in accuracy forecast). So I do think that might be the connection the authors are missing as the core idea of the paper - that MSE is everything

More deeply, I am not sure any decision making algorithm would actually care about the diffusion coefficient of an SDE

**Main Review:**

Though the paper tracks really interesting metrics. I think it still misses the point of what the martingale property really means. In Foster and Stine, the core idea seems to be that the martingale filter allows you to have a *more* accurate forecast. That means any other metric but accuracy (so straight MSE) doesn't really matter - as the martingale filter illustrates. In that context, Im not sure what this paper adds. That seems to be the crux of Eisenach et al too - the decoder self allows the MQNet to be more accurate.

A few clarifications -

1) Do you assume your forecasts are already calibrated? If so, do you ensure this for all the models (specially MMFE before you compare)
2) Is MQLSTM a Bayesian model?
3) MQLSTM seems the wrong baseline - MQ_CNN_Wave within the paper is more accurate (seems to just be a wavenet encoder instead of an LSTM encoder) but even then Eisenach et al (MQTransformer) actually tries to make the MQLSTM a martingale through the decoder self attention

**Time Spent Reviewing:**

2

---

> ### Author Response · Authors · 2021-08-10
> **Response to reviewer's questions**
>
> We appreciate the thoughtful and thorough review. We agree with Foster and Stine that methods that output forecasts that satisfy the martingale property often lead to better accuracy, as measured by MSE. However, one important clarification is that we believe the martingale property itself is better viewed as ensuring models are internally consistent, rather than solely as a means to achieve better accuracy. In particular, as Foster and Stine do with their proposed diagnostics, one can evaluate the extent to which forecasts satisfy the martingale property separately from the accuracy of the underlying model.
>
> Regarding the specific clarification questions that were posed:
>
> 1. No, we don’t assume the forecasts in the training data are calibrated. In order to apply our inference procedure, we only require that the forecasts are between 0 and 1 at each time point. Proposition 1 then guarantees the estimated GLIM model satisfies the martingale property. Nonetheless, in our experiments, we use first-order models that produce approximately calibrated forecasts (see Figure 8 in the supplementary material), as better first-order models (e.g., those that are calibrated) will generally lead to better downstream inferences.
>
> 2. MQLSTM itself is not a Bayesian model. However, we used dropout in the LSTM layer to approximate the posterior distribution, as suggested by Srivastava et al. (2014).
>
> 3. We agree that there are many different baseline models one could compare against. As noted above, we view the primary contribution of our work as describing a principled way of constructing forecasts that satisfy the martingale property (and, in turn, are internally consistent), rather than solely seeking to achieve the lowest MSE.

---

> > ### Comment · Reviewer_j4Fr · 2021-08-11
> > **Response**
> >
> > Hi All,
> >
> > Thanks for the thoughtful response. But I think I cant agree. Going through the claims:
> >
> > 1) You say "Finally, GLIM generates predicted paths that are both calibrated (as is theoretically guaranteed)". What is the proof here? Just satisfying the martingale property does not guarantee calibration - as a constant forecast will be trivially a martingale but may have the wrong level (event occurs 10% of the time, a constant forecast of 30% is a martingale but not calibrated).
> >
> > 2) I think any Neural Network trained to Quantile Regression should resemble empirical bayes (https://tinyurl.com/ps4mut58). So Im not sure I think this is correct. What would be a proof here?
> >
> > 3) I interpreted your claim as martingality is the win. So to the extent that you agree with Foster and Stine - maybe another way to put what Im saying is that utilizing a forecast for a decision is the primary purpose of a forecast - and whether a forecast is a martingale or not, is only relevant to the extent that there exists an "embarrassing" error in the forecast itself! Namely, there is a trivially more accurate forecast that can easily be constructed from a non-martingale forecast that is more accurate.
> >
> > The analogy is calibration. An uncalibrated forecast is not "wrong" in any sense beyond that there exists a trivial way to construct a more accurate forecast (i.e. by calibrating it).
> >
> > So Im not sure a full SOTA accuracy comparison isnt warranted! The result of Foster and Stine seems to deeply be - a lack of martingality is only relevant till the existence of a trivially more accurate forecast. So the regression they have is a simple procedure for making any forecast more accurate. They offer a way to compare the signal extraction abilities of different models by making them all martingales (and thus removing idiosyncracies of the model and focusing on its signal extraction capabilities).

---

> > > ### Author Response · Authors · 2021-08-12
> > > **Response**
> > >
> > > Thank you for your prompt response and please see below for our replies:
> > >
> > > 1. GLIM describes a data--generating process from which the sampled paths are *internally* calibrated, by which we mean that for the path distribution output from a GLIM model with arbitrary parameters, we have, for t > s, E[Y_t | Y_s  = p] = p. This property follows directly from Proposition 1. In particular, Pr[Y_T = 1 | Y_s = p] = p, where Y_T is the final outcome at time T inferred under the model. We graphically show this pattern for an illustrative example in Figure 2 for GLIM, and show it numerically for GLIM and our baseline models in Figure 3(a-b).
> > > If, however, the first-order prediction at time Y_s is not itself calibrated with respect to the observed outcomes, then the GLIM predictions will also fail to be *externally* calibrated. As noted in our previous response, we take care to ensure our first-order models are in fact calibrated, as shown in Figure 8 in Appendix E.
> > >
> > >
> > > 2. Yes, you are correct, and we apologize for the confusion we introduced. Our previous response inadvertently referred to a Bayesian LSTM baseline that we used in an earlier version of our paper (for which dropout was used to generate posterior estimates). In the submitted draft, we ultimately replaced Bayesian LSTM with the more recent MQLSTM, as it gave stronger baseline results. To avoid further confusion, we will release our code upon publication for cross-checking.
> > >
> > >
> > > 3. We agree that MSE is an important measure of performance in many applications, but we also believe that it is not the sole criterion of model quality in all situations. As noted in the statistics and economics literature (see, for example, Augenblick and Rabin (2021), Gelman et al. (2020), Taleb et al. (2019), Taleb (2018), and Ely et al. (2015)), the martingale property is often important to ensure predictions can be interpreted as “true” probabilistic forecasts that satisfy the basic laws of probability. In particular, the martingale property is fundamental to modern asset pricing as a means to guard against arbitrage. While it may be possible to optimize for some decision-centric utility functions without probabilistic forecasts, these “unfaithful” probabilistic forecasts are often hard to reason about. Our claim is not that the martingale property is necessary in all applications, but that it is often useful to consider, and our work attempts to shed light on this understudied area of time-series analysis.

---

### Official Review · Reviewer_DdpC · 2021-07-21

**Rating:** 7
**Confidence:** 4

**Summary:**

A new model for the evolution of predictions with strong empirical results.

**Ethical Concerns:**

None.

**Limitations And Societal Impact:**

Yes.

**Main Review:**

This paper considers the problem of estimating the probability of a future binary outcome. The authors look at the question of how a prediction evolves over time as new information is made available. They develop a Bayesian time series model to look at the structure of probabilistic forecasts over time. A salient feature of such forecasts is a martingale property, viz., the expected value of future predictions along the forecast path is equal to the expected value of the current prediction. This effectively constrains the forecasts and puts the focus on the study of conditional distributions of paths given the information at the forecast time. The authors perform a good review of work in modelling time series and point out the special structure in probability paths that needs to be captured and is not performed by some existing model classes.

The paper then moves on to presenting the main model, the GLIM. This is a kind of Gaussian latent variable model, which models the evolution of the probability of the binary outcome by adding `positive' or `negative' information incrementally over time. By conditioning on information previously added, the certainty of the prediction increases as the event of interest is approached. A simple and clear inference process is described for the model allowing computing the likelihood by first applying a recursion to update the means and variances of the Gaussians in the latent space and then computing an overall likelihood of the probability path. Since this is a novel model, I would have appreciated some more intuition on the interpretations of these recursions, ie,. giving a bit more meaning to the parameters that are being computed.

Finally, two well-chosen data sets (basketball and weather) are used to test the model. While the description of the model and inference process is very clear, the experiments section is harder to follow. In part, I think this is due to the novelty of the model. You should clearly outline the experiment you are doing and use the appropriate notation. Thus, for example, how exactly do you fit your model to the basketball data?

I understood that you take a set of basketball games, say 1 , ... , N. You compute a probability path for each game, {y_1^1 , ... , y_T^1} , ... , {y_1^N , ... , y_T^N}. Then you take these paths and plug them into the procedure described by Theorem 1. This gives you an estimate of the covariance parameters for the structure you described in lines 227-228. You allude to a prior on these parameters that is `weakly informative' but do not specify it anywhere. You then test the model on a single game given (1) a prediction at the 24th minute, presumably from the same sort of random forest model, and (2) the parameters inferred on the entire set of data available. Is this single game part of a test set? Finally, a general comment is why do you do your inference by drawing a single value from the posterior, wouldn't it make more sense to use the posterior mean or even the maximum likelihood estimate for that matter? Can you say something about the posterior of theta?

Overall, I think this paper is certainly an accept, the model is novel, promising, and supported by good empirical results.

**Time Spent Reviewing:**

3

---

### Decision · Program_Chairs · 2021-09-28

**Decision:**

Accept (Poster)

**Comment:**

After extensive discussions that led to reviewers rethinking their opinions significantly, there was a consensus that this paper, though well-written and proposing an interesting and novel model, is not quite ready for publication. The core concern is that martingality in and of itself is not really an end goal -- and to the extent that it is, the Foster & Stine approach can be used to post-process martingality onto a model. Thus, to really demonstrate the value of the proposed approach, it seems necessary to at least understand its accuracy (say, MSE) in comparison to that simple baseline. Please see the updated reviews for additional details.

**Consistency Experiment:**

NeurIPS has a long history of experimentation. In 2014, NeurIPS ran an experiment in which 10% of submissions were reviewed by two independent committees to quantify the randomness in the review process. This year, we repeated a variant of this experiment to see how the quality of the review process has changed over time.  This paper was part of the experiment and was therefore assigned to two committees (consisting of reviewers, an Area Chair, and a Senior Area Chair) that reached independent decisions.  If both committees made the same recommendation, this recommendation was followed. If a single committee recommended acceptance, the paper was accepted (with the exception of a few cases in which the other committee identified what we considered a fatal flaw, e.g., an error in a key result).

This copy’s committee reached the following decision: **Reject**

The other committee assigned to the paper recommended **Accept (Poster)**.  You can find the other set of reviews, along with any follow up discussion with the authors here:
https://openreview.net/forum?id=OU4LL1qP3Dg